# Evolution of a Superhydrophobic H59 Brass Surface by Using Laser Texturing via Post Thermal Annealing

**DOI:** 10.3390/mi11121057

**Published:** 2020-11-29

**Authors:** Xizhao Lu, Lei Kang, Binggong Yan, Tingping Lei, Gaofeng Zheng, Haihe Xie, Jingjing Sun, Kaiyong Jiang

**Affiliations:** 1College of Mechanical Engineering and Automation, Huaqiao University, Xiamen 361021, China; kangleihqu@163.com (L.K.); qqyanrui@163.com (B.Y.); tplei@hqu.edu.cn (T.L.); 2Department of Electronic Science, Xiamen University, Xiamen 361005, China; 3Fujian Key Laboratory of Special Energy Manufacturing, Huaqiao University, Xiamen 361021, China; 4Department of Mechanical Engineering, Xiamen University, Xiamen 361005, China; zheng_gf@xmu.edu.cn; 5Department of Mechanical Engineering, Putian University, Putian 361110, China; haihexie@163.com; 6School of Materials Science and Engineering, Xiamen University of Technology, Xiamen 361024, China; 2015000093@xmut.edu.cn

**Keywords:** superhydrophobic, thermal annealing, laser texture, muffle furnace, nanoparticles

## Abstract

To fabricate an industrial and highly efficient super-hydrophobic brass surface, annealed H59 brass samples have here been textured by using a 1064 nm wavelength nanosecond fiber laser. The effects of different laser parameters (such as laser fluence, scanning speed, and repetition frequency), on the translation to super-hydrophobic surfaces, have been of special interest to study. As a result of these studies, hydrophobic properties, with larger water contact angles (WCA), were observed to appear faster than for samples that had not been heat-treated (after an evolution time of 4 days). This wettability transition, as well as the evolution of surface texture and nanograins, were caused by thermal annealing treatments, in combination with laser texturing. At first, the H59 brass samples were annealed in a Muffle furnace at temperatures of 350 °C, 600 °C, and 800 °C. As a result of these treatments, there were rapid formations of coarse surface morphologies, containing particles of both micro/nano-level dimensions, as well as enlarged distances between the laser-induced grooves. A large number of nanograins were formed on the brass metal surfaces, onto which an increased number of exceedingly small nanoparticles were attached. This combination of fine nanoparticles, with a scattered distribution of nanograins, created a hierarchic Lotus leaf-like morphology containing both micro-and nanostructured material (i.e., micro/nanostructured material). Furthermore, the distances between the nano-clusters and the size of nano-grains were observed, analyzed, and strongly coupled to the wettability transition time. Hence, the formation and evolution of functional groups on the brass surfaces were influenced by the micro/nanostructure formations on the surfaces. As a direct consequence, the surface energies became reduced, which affected the speed of the wettability transition—which became enhanced. The micro/nanostructures on the H59 brass surfaces were analyzed by using Field Emission Scanning Electron Microscopy (FESEM). The chemical compositions of these surfaces were characterized by using an Energy Dispersive Analysis System (EDS). In addition to the wettability, the surface energy was thereby analyzed with respect to the different surface micro/nanostructures as well as to the roughness characteristics. This study has provided a facile method (with an experimental proof thereof) by which it is possible to construct textured H59 brass surfaces with tunable wetting behaviors. It is also expected that these results will effectively extend the industrial applications of brass material.

## 1. Introduction

Today, an anti-corrosion related technology based on brass has increased many people’s interest in mimicking the pattern of the lotus leaf surface. The reason is that a nanosecond fiber laser can be used to create this type of pattern on an H59 brass surface, thereby forming super-hydrophobic surfaces after an incubation period of about 2 weeks. This means that an increase in water contact angle (WCA) of up to 150° can be obtained after about 14 days of evolution [1,2,3]. However, the possibility of producing a superhydrophobic H59 brass surface in less than 14 days remains a big challenge. While annealing the brass material, the temperature will continuously increase, and the laser-induced texture will become both broader and wider. This process will simultaneously reduce the surface energy, and speed up the super-hydrophobic properties of the brass surface.

As an important engineering metal material, H59 brass has become widely used in various types of applications (in industrial hydraulic systems, for transportation, in ship propellers, etc.). For this brass, a quick transformation from hydrophilicity to hydrophobicity should effectively improve the anti-corrosion properties. Examples of potential applications for this type of modified H59 brass material are self-cleaning surfaces [4,5], oil/water separations [6], anti-icing/frosting [7,8], drag reduction [9], and microfluidic devices [10]. As a result of recent research efforts, appropriate brass surfaces with hierarchic micro/nanostructures (from here on called micro/nanostructures), containing nanosized grains and Cu_2_O nanoclusters, have been manufactured with the purpose of producing hydrophobic surfaces with very short evolution times. These hydrophobic surfaces have been formed by using various types of deposition techniques; the sol–gel method, chemical vapor deposition [11,12], chemical etching [13,14], nanoimprint lithography [15], and self-assembly [16]. However, these methods are limited in terms of high cost, complicated processes, and poor mechanical properties of the textured surfaces. Recently, some environmentally friendly chemical solutions (like isopropyl alcohol, sodium bicarbonate solution, etc.) have been used to shorten the wettability transition time (down to about 5 days) [17]. In recent years, different types of annealing treatments, at lower temperatures, have been used to speed up the super-hydrophobic wettability transition [18]. The surface energy, roughness, and morphology, as well as the dual-scaled particles in the surface micro/nanostructure, play important roles in this evolution procedure (with an exception for the Cu_2_O nanoclusters) [19,20]. As a matter of fact, the physical profile of the hierarchic micro/nanostructures, in addition to the large oxygen content, will speed up the variation of Cu_2_O material on the surface [21,22].

It is well known that the super-hydrophobic property of a surface is determined by a combination of physical roughness and chemical composition. In fact, super-hydrophobic surfaces do not only depend on dual micro/nanostructures, but also on the existence of low surface energy Hence, key to the evolution of Cu_2_O during the laser processing of the brass surfaces is the reversible wettability induced by different laser parameters [13]. Different annealing treatments will also induce phase transitions between the alpha and beta phases of brass, with different metallographic sizes. Furthermore, the different WCAs are determined by the different nanograin sizes, which in turn depend on the different hierarchies in the micro/nanostructure (as manufactured by the combination of laser processing and Cu_2_O evolution). 

## 2. Setup and Experiments

### 2.1. Experimental Setup 

A fiber laser (PicoYL-40-600-20, Yangtze Soton Laser Co., Ltd., Wuhan, China), with a wavelength of 1064 nm, 0.6–4 ns Pulse duration was used for the texturing of different micro/nanostructures on heated brass surfaces. After annealing in a Muffle furnace, the sample surfaces were polished with silicon carbide abrasive papers (500#, 800#, and 1200#, YING QIU, China). The micro/nanostructures on the brass surfaces were characterized by using a field emission scanning electron microscope (FESEM) (Zeiss SUPRA 55, Munich, Germany). Finally, the chemical compositions were determined by using energy-dispersive X-ray spectroscopy (EDS). Moreover, the water contact angles were measured to evaluate the surface wettability transition, and this was performed by using a WCA analyzer (JC2000D, Shanghai Zhong Chen Digital Co., Shanghai, China), and the sessile drop technique. The main experimental setup, including all individual characterization and processing steps, is demonstrated as follows: The polished samples after annealing heat treatment were successively immersed in isotone and alcohol liquids, where they were ultrasonically cleaned for 30 min each. The samples were thereafter kept in air and grated with a nanosecond fiber laser. This process brought about the evolutionary circumstances that lead to the different superhydrophobic surfaces.

### 2.2. Annealing of H59 Brass Substrates Prior to Laser Texturing

The brass samples were heated in a Muffle furnace up to the brass phase transition temperatures of 350 °C, 600 °C, and 800 °C. The samples remained at these temperatures for about 2 h and thereafter cooled in air. The roughness of each sample surface was observed to increase with an increase in temperature. In addition, the content of the metal and metal solid solution, in the brass, became more pronounced with a higher annealing temperature (see Table 1). When the brass was annealed to 350 °C, the main phase of the brass was the alpha phase. After annealing to 600 °C, the metallic alpha and beta phases did coexist. When heating the sample above 800 °C, an amorphous state was formed [23,24,25]. In addition, the collection of nanograins became large. The heated brass substrates showed a small reflection index with deeper grooves, and the distances between the nano-clusters were enlarged. The contact area between a water drop and this type of coarse surface was quite small, which was observed to enhance the speed for reversing the super hydrophilicity to super-hydrophobicity.

As a result of thermally annealing, different types of metallographic were observed on the H59 brass substrate. Those surface structures are shown in Table 2 [22,23,24]. Laser grating was thereafter used to form dual hierarchy micro/nanostructures on the brass surfaces (i.e., nano-grains evolution on the micro-grooves).

In the present study, the areas of micro/nanostructured hierarchic morphologies were chosen from different parts of the sample surfaces, which were due to the appearance of different phases. After the optimization of the laser parameters (see Figure 1), characterizations by FESEM were made to obtain information about the surface morphologies of the fabricated surfaces of the brass samples. In addition, WCA measurements were performed to obtain information about the wettability transitions. The evolution mainly implied the growth of chemical composition, mainly depended on the morphologies of laser processing. The Silicon carbide abrasive papers (500#, 800#, and 1200#) had been used in polishing the different brass samples (after the annealing processes), the reflection indexes of the surfaces were found to be solely dependent on the different nano tissues.

Table 2 shows the laser texturing parameters that were used in texturing the first row of samples in Table 1. As can be seen in Table 2, the laser fluence was tuned to 7.16 J/cm^2^, 8.60 J/cm^2^, 10.02 J/cm^2^, 11.46 J/cm^2^, and 12.88 J/cm^2^. Moreover, the waist of the laser spot was about 20 μm, while the laser scan speed, period, and repetition rate were set to 1 mm/s, 100 μm, and 200 kHz, respectively. These laser fluences have the capacity to change both the brass roughness and brass oxidation [21]. When scanning the grating with a higher laser fluence, deeper grooves will be dug. At the same time, laser energy beyond the brass threshold will enhance the oxidation on the surface. As is clear from Table 1, a higher energy density grating will enhance surface roughness, and increase surface oxidation. A deeper groove means that there is a larger surface area for oxidation to take place. For this type of surface, wettability will be able to go from hydrophilicity to hydrophobicity in a very short time. 

For the second row of samples in Table 1, the following parameter values have been used: 80 μm, 90 μm,100 μm,110 μm,120 μm (laser grating period), 1 mm/s (laser scan speed), 8.60 J/cm^2^ (laser fluence), and 200 kHz (repetition rate). According to the Cassie–Baxter formulation, a microstructure grate will be formed by the laser. The dimension of the microstructure is about 100 μm, which will enhance the surface roughness in accordance with the increased laser fluence. The chosen size has been experimentally discovered to be reasonable. For the third row on the samples in the Table 1, the laser scan speeds were tuned to 0.1 mm/s, 0.5 mm/s, 5 mm/s, 10 mm/s, and 100 mm/s, respectively. The period, laser fluence, and repetition rate were kept fixed (100 μm, 8.60 J/cm^2^, and 200 kHz, respectively). Finally, for the samples in the last row in Table 1, the laser repetition rates were 100 kHz, 500 kHz, 1000 kHz, 2000 kHz, and 5000 kHz, with the constant period, laser fluence, and scan speed (100 μm, 11.46 J/cm^2^, and 5 mm/s, respectively). 

The WCAs were measured after laser processing. In addition, the laser parameters were also optimized. From the resulting graphs in Figure 1, Figure 2 and Figure 3, including information about relevant nanoparticles, nanograins, surface roughness, and wettability transition, the following important information can be understood. After the annealing processes, nanoparticles became attached to, and even in some cases included in, the different nanograins. In addition, there was a lesser amount of nanograins on the sample surface. Furthermore, the texturing on the brass surface became more pronounced. However, the smaller dimensions of the micro/nanostructures mean that there is a much larger area where the formation of compounds like CuO and Cu_2_O could take place.

## 3. Evolution of WCAs on the Brass Sample Surfaces 

The evolution of WCAs was observed for 2 weeks, and statistical analysis was conducted. The laser parameters (like laser fluence, scan speed, repetition rate, and period) were also optimized by using the different micro/nanostructures as quality indicators. To be more specific, these parameters relate to the different micro-configurations, nanosized grains, nanoparticles, etc. The nanograins and nanoparticles that were produced by the nanosecond fiber laser were relatively coarse. However, this was not the case for other types of nanostructures. The different laser parameters therefore had to be optimized.

When a water-drop falls to the textured surface, the most prominent part of the hierarchic surface will be the first to be in contact with the water. According to the Wenzel formula, when the surface roughness increases, the contact area between the surface and the water droplet decreases. As a result, the intensity of the contact pressure becomes enlarged, and the water drop must overcome a much larger repellant.

On the other hand, the surface energy will decrease and the water contact angle will quickly become quite large. In addition, the surface attractive forces toward the liquid drop will become smaller for the polished surface. When the sizes of the nanograins become smaller, the distances between the nanoclusters will be longer. This is the result of a transformation of the metal phase. Hence, Cu_2_O nanoclusters can be formed quite quickly. The air will thereafter become attached to the surface as a cushion-like structure; nanograins on the surfaces resemble and are looked like American Acacia when the temperature is sufficiently high. According to the Cassie–Baxter formula, the distances between the clusters of nanograins at 600 °C and 800 °C are much longer than those at 350 °C. This will result in a decrease in nanograin density, while the nanoparticles will become larger, and the surface roughness is more pronounced. The gap between nanoclusters will be decreased, the oxygen breath of brass will be blocked and Cu_2_O will be halted.

### 3.1. Dependence on WCA Evolution by Laser Fluence and Annealing Temperature

Figure 1 shows the evolution in WCAs when going from hydrophilicity (with a contact angle smaller than 90°) to hydrophobicity (with a contact angle larger than 90°). Brass surfaces have been textured after being annealed at different temperatures. As can be seen in Figure 1, the laser impact on the water contact angle is large. For instance, the laser fluence will determine the depth of the grooves. There is also a directly proportional relationship between the laser power and the laser fluence, in that low laser power is directly correlated with low laser fluence. As a result, a low laser fluence relates to the processing of plain grooves. Moreover, the WCA will increase slowly for surface texturing with a low laser fluence, in comparison to the corresponding treatment with high laser fluence. When the polished brass mesh (as textured by the laser process) was very shallow, the WCA could remain through the micro/nanostructure evolution. As the processed depth increased, there was a larger volume of processed brass material removed from the grooves (which was due to an increased laser fluence). The laser fluence affected the sizes of both the grooves and the grains in the hierarchic micro/nanostructure on the brass surface.

As a first measurement, water drops were pumped to a textured brass surface. This low-energy surface was expected to form a smaller amount of Cu_2_O and to also enlarge its WCA. While the WCA was measured in the first measurement, a textured brass surface, with deeper grooves, was processed by using a higher laser fluence. Since the higher laser fluence was an indication of a higher laser single pulse energy, which excavated a broader and deeper micro groove size, further oxidation on the surface did also take place [26]. As a result, a higher proportion of CuO will be manufactured on the surface. Besides, there was an increase in attractive forces to this roughened surface area. The gap is big; therefore, the proportion of oxygen will run away. As a result, the surface showed much higher surface energy and developed initial hydrophilicity. Most of the surface will be endowed with large hydrophilicity attraction; after all, C-O will release its oxygen because of the large gap between hierarchy structure, and the transformation to hydrophobicity will take place more quickly. 

Because of the existence of surface oxidation, the significant hydrophilic effect was shown already in the first measurement. When the laser fluence increased, the large distances between the nanograins made the WCA increased very fast. The hierarchic nanostructures, produced at different annealing temperatures, can be seen in Figure 1f–h. After an incubation time of two weeks, the low-temperature annealed brass sample had developed a dense nanograin formation (also including nanograin boundaries). 

The coarseness of the nanograins made the surface energy to increase quite slowly. On the contrary, the high-temperature annealed brass sample showed small nanograins (see Figure 1f–h). The WCA increased very quickly for this type of surface. The large distances between the nanograins could easily become filled with, for example, air. According to the Cassie–Baxter formula, water drops in contact with a large volume of air will have a large capacity to reduce the surface energy (which implicates a fast increase in the WCA). The developing speed of the WCAs for the 800 °C annealed sample was found to be higher than the developing speed for the 350 °C annealed sample.

### 3.2. Dependence on WCA Evolution by Scan Periods and Dual Hierarchy Micro/Nano-Structures

According to the Cassie–Baxter and Wenzel formula, the dimension of the surface microstructure is related to the surface roughness. A laser scanned the annealed surface (with 100 μm periods) will enhance the annealed brass surface roughness. 

According to the result of the experimental measurements, and related references [27,28,29,30], the microgroove and dual microstructure acted as an airbag while the droplet fell back to the roughened surface. A fiber laser spot’s diameter was arranged in continuously linear grooves. Moreover, the studies of microstructure related to super-hydrophobicity were concentrated on some specific sizes. The experimental scan speed was set to 1mm/s. While the dual micro/nanostructure enlarges the contact area with oxygen, the nanostructure will speed up the cuprous oxide’s evolution in the groove.

As can be seen from this experiment, there was a stable influence by the micro-groove on the wettability (Figure 2); on the contrary, the laser-textured roughness was apparently affected by thermal annealing processing. The high temperature is most probably the major cause of the superhydrophobic surfaces in Figure 2f–h. 

From these experiments, optimal periods were chosen, for which microstructure morphologies were further analyzed. The formed surface roughness, including the temperature factors, was found to have a certain impact on the surface properties.

### 3.3. Dependence on WCA Evolution by Scan Speed and Annealing Temperature

When the laser scanned the annealed surface at a certain high speed, the laser spots were not arranged in continuously linear grooves. The studied hot spots were instead concentrated in some specific locations. The scan speed was also observed to have a major influence on both the accumulated temperature and the metal oxidation, while the WCAs on the brass surfaces did not differ from each other. 

When the laser scanning speed is lower, the laser beam stays longer on a certain area on the surface of the brass. Moreover, the larger the nanoparticles on the surface, the easier it will be for the temperature to increase. However, this is not the situation for the WCA, which will not be easy to increase. According to the Cassie–Baxter formula, the more hydrophilic the nanograins are, the more hydrophobic they will become. Moreover, the smaller the micro-groove spacings, the greater the surface roughness. In addition, the large oxidation degree of the surface can be seen by the color of the brass surface. The black and dark colors show a high oxidation concentration [31,32]. In the initial tests, large surface energy was connected to a more hydrophilic surface [33,34,35]. In addition, the increase in annealing temperature was found to have a significant effect on both the increase in the number of nanograins and of the hydrophobicity [36,37,38].

### 3.4. Dependence on WCA Evolution by Repetition Rate and Annealing Temperature

The effect of fabrication on the laser repetition frequency, in relation to the WCA and micro/nanostructures, was also significant. The higher the laser repetition rate, the smaller the single pulse energy; in addition, the roughness will be lower than the others, and the micro-grooves shallower. It was even difficult, and sometimes not possible, to fabricate the necessary microstructures. Furthermore, the higher the laser frequency, the less likely it was to affect the results after heat treatment. In addition, the higher the heat treatment temperature, the more hydrophilic the surface. Due to the increase in laser repetition frequency, and the decrease in single pulse energy, the copper oxide was shown to be less affected by the heat of the laser processing.

Therefore, the WCA of the brass surface that has been heat-treated at 350 ℃, became larger than the WCAs for 600 °C and 800 °C. 

Figure 4 shows the revolution of WCA because of laser repetition rate, on the other hand, Figure 5 reveals the existence of nanograins in brass samples that have been textured after different thermal annealing treatments. The CuO nanograins were expected to increase during the higher temperature annealing treatments. In addition, the distance between the nano-clusters became enlarged. As a result, the oxygen concentration on the surface increased, which was mainly due to the huge free space on the surface. The surface Cu atoms could thereby bind to gaseous O_2_ until they reached their solid solution concentration. Those would lead to a small surface energy and the improvement of water resistance ability.

## 4. Results

The different annealed samples have been textured by using a nanosecond infrared fiber laser. The samples have thereafter been optimized with the purpose to speed up the transition wettability (from a hydrophilic surface to a hydrophobic surface). By using a higher fluence of the laser, deeper and broader grooves in the brass could be formed. Moreover, the larger the annealing temperature, the higher the mechanical properties of the brass. In addition, the larger the distances between the nanograins, the greater the surface micro/nanostructure roughness. Moreover, the smaller the nanoparticles, the finer the nanostructure (which will make it easier to form a good solid–liquid contact). When the material itself is hydrophobic, a large degree of pronounced surface roughness will induce a large degree of hydrophobicity on the contact surface area. 

As can be seen in Figure 6b, d and f, when annealing the sample to 350 °C, the nanostructure became finer, and the number of nanograins was smaller and more scattered (as compared with the results of the other annealing temperatures). In short, the sizes of the nanoparticles did relate to the temperatures of annealing. Moreover, the sizes of the Cu_2_O clusters were found to be similar to the size of the nanograins at an increased annealing temperature. As a result, the surface roughness became related to the microstructure.

During the initial tests, when the laser energy was enhanced, the degree of hydrophilicity also increased. It can also be concluded from the measurements that were made after the annealing processing, that a higher annealing temperature is connected to a stronger resistance towards laser processing on the brass sample surface. Directly after the annealing of the surfaces, a larger laser fluence was also observed to create a smaller WCA (i.e., the degree of surface hydrophilicity increased). However, this changed after 4 days of evolution, when the super-hydrophobicity grew stronger.

As can be seen in Figure 7b,d,f, it is quite obvious that the nanograins became smaller in the 350 °C annealed sample, as compared with the results for 600 °C and 800 °C annealed samples. Higher temperatures resulted in micro-grooves with narrower nanograins. On the other hand, Figure 7b,f,i show that the nanograin sizes will be reduced when increasing the annealing temperature. An evolution of the Cu_2_O nanostructure was also formed, which is shown in Table 3.

According to the results obtained by Energy Dispersive Analysis Spectroscopy (EDS), the composition of the samples could be identified. In addition, the WCA evolutions were measured with confidence. There were no other impurities sub-quenched into the sample substrates.

## 5. Conclusions

Different annealing treatments of the brass samples have resulted in different wettability transition times, which in turn were caused by the existence of different amounts and types of nanoparticles, nanograins, and grain boundaries. In addition, the hierarchic micro/nanostructures played an important role during the reversible transition from super-hydrophilicity to super-hydrophobicity.

(1)The different distances between the nanograins were observed to affect (a) the reversible wettability time, (b) the nanoparticles, that in turn were related to the progress of the Cu_2_O evolution, and (c) the surface roughness (that would influence the surface energy). After the initial nanosecond laser ablation, the high laser fluence did increase the distances between the nanograins. More CuO materials were formed, and the surfaces received super-hydrophilic properties. The CuO composition infiltrated the grooves and became more easily embedded into the ablated microstructures. After the evolution of Cu_2_O, the surfaces were transferred to super-hydrophobic properties. To reduce the laser repeat repetition, it interacts with the H59 brass surface with a higher power in each pulse, and then, the brass oxidation was extended and the distances between the nanograins were increased. Furthermore, while the laser rate was turned down, the laser was removing an increased volume out of the micro-grooves. All of these methods were found to speed up the evolution of Cu_2_O and WCA.(2)The annealing temperatures were strongly coupled with the evolution of nanoparticles. High temperatures induced much smaller nanoparticles and increased the adhesion of Cu_2_O to these nanoparticles. Moreover, the number of nanoparticles increased rapidly, which was also the situation with the WCAs.(3)The chemical properties of brass played a fundamental role in the evolution of super-hydrophobicity. For instance, the hydrophilic properties of the brass surfaces were related to the number of CuO nanoclusters. The transformation from a hydrophilic to a hydrophobic surface was then related to the existence of Cu_2_O nanoclusters in the micro/nanostructures, and to the surface energies.

The more pronounced micro/nanocomposite structures were more easily adsorbed by ions and formed a type of air cushion attached to it. This resulted in liquid–solid bubbles and formed a gas–liquid–solid composite in contact with the surface. With an increase in surface roughness, the solid–liquid contact area did decrease, thereby forming a hydrophobic surface. It has here been assumed that the droplet will be in direct contact with the solid surface, with no air in between. The contact angle increased, and the surface became more hydrophobic. Hence, when the solid surface is made of a hydrophilic material, the increased surface roughness will cause a more hydrophilic surface.

## Figures and Tables

**Figure 1 micromachines-11-01057-f001:**
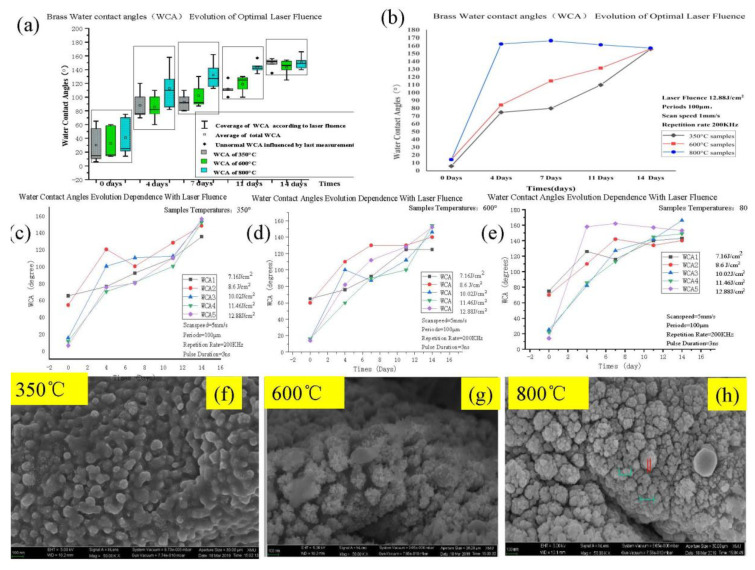
(**a**) shows the dependence of the WCA evolution on the different annealing treatments and laser fluence. The coverage of different laser fluence included 7.16 J/cm^2^, 8.6 J/cm^2^, 10.02 J/cm^2^, 11.46 J/cm^2^, 12.88 J/cm^2^ in (**a**), which is included in Table 2; the average WCA can almost stand the hydrophobic wettability, the abnormal WCA is influenced in terms of the waterdrop contact during the last measurement. (**b**) shows the different wettability transition speeds after annealing and by using various laser fluences. The diagrams of curves show the WCA transitions for laser-textured brass samples that have been annealed at (**c**) 350 °C, (**d**) 600 °C, and (**e**) 800 °C (being affected by different laser scan speeds). (**f**) a 350 °C annealed brass sample with larger nanograins (and finer metallic texture), as compared with the samples in (**g**,**h**). (**g**) the beta phase of brass with a coarse metallic texture, as compared with the result shown in (**d**).

**Figure 2 micromachines-11-01057-f002:**
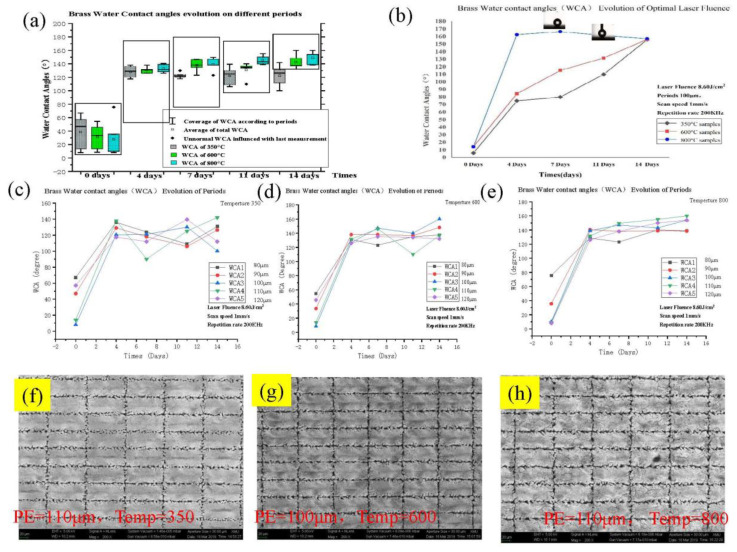
(**a**) shows the dependence of the WCA evolution on the different annealing treatments and scan periods. The coverage of WCA refers to the different grating periods, which are 80 μm, 90 μm, 100 μm, 110 μm, 120 μm in Table 2; the average WCA almost reveals the hydrophobic wettability, the abnormal WCA is influenced in terms of the waterdrop contact during the last measurement. (**b**) shows the different wettability transition speeds after annealing and by using various periods. The diagrams of curves show the water contact angle transition for (**c**) 350 °C, (**d**) 600 °C, and (**e**) 800 °C annealed laser-textured brass surfaces (being affected by different laser scan periods). Surface morphologies of laser-textured and annealed samples at (**f**) 350 °C, (**g**) 600 °C, and (**h**) 800 °C.

**Figure 3 micromachines-11-01057-f003:**
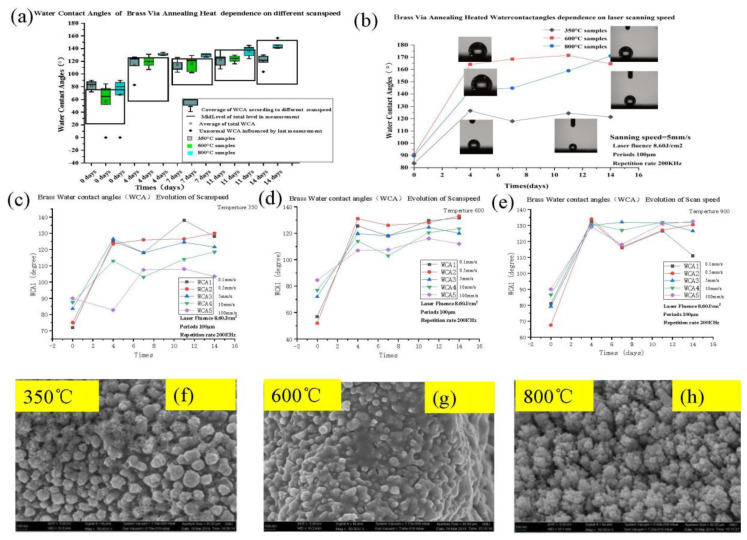
(**a**) dependence of the WCA evolution on the different annealing treatments and scan speed. The coverage of WCA refers to the different scan speeds were tuned between 0.1 mm/s, 0.5 mm/s, 5 mm/s, 10 mm/s, 100 mm/s in Table 2; the average WCA can almost stand the hydrophobic wettability, the abnormal WCA is influenced in terms of the waterdrop contact during the last measurement. (**b**) different wettability transition speeds after annealing and by using various speeds. The diagrams of curves show the water contact angle transition for (**c**) 350 °C, (**d**) 600 °C, and (**e**) 800 °C annealed laser-textured brass surfaces (being affected by different laser scan speeds). Surface morphologies of laser-textured and annealed samples at (**f**) 350 °C, (**g**) 600 °C, and (**h**) 800 °C.

**Figure 4 micromachines-11-01057-f004:**
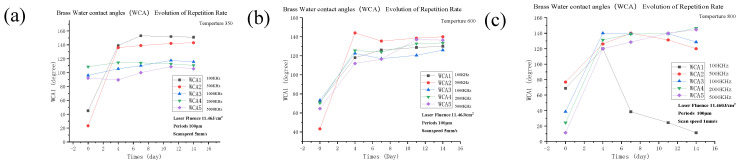
The graphs showing the WCA transition at annealing temperatures of (**a**) 350 °C, (**b**) 600 °C, and (**c**) 800 °C. It is also shown that the brass surfaces are affected by different repetition rates.

**Figure 5 micromachines-11-01057-f005:**
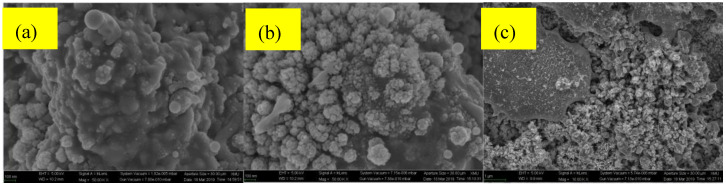
Morphologies of laser-textured and (**a**) 350 °C (scale bar 100 nm), (**b**) 600 °C (scale bar 100 nm), and (**c**) 800 °C annealed samples (scale bar 1000 nm).

**Figure 6 micromachines-11-01057-f006:**
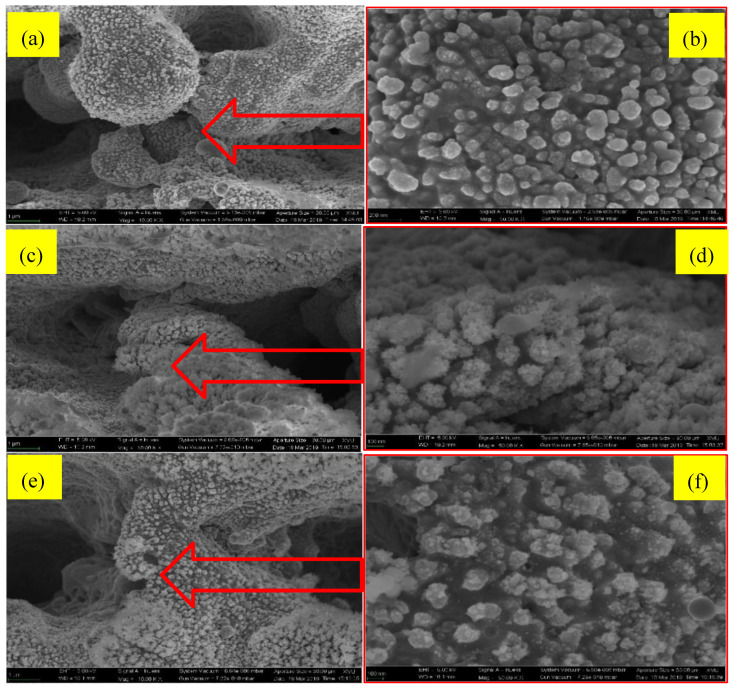
FESEM pictures showing the micro/nanostructure morphologies of hydrophobic surfaces that have been textured by using a 4.18 J/cm^2^ laser fluence, in addition to the following annealing temperatures: (**a**,**b**) 350 °C, (**c**,**d**) 600 °C, (**e**,**f**) 800 °C, the scale bar of the SEM Figure in the left row is 1 micrometer, the right row is 100 nm.

**Figure 7 micromachines-11-01057-f007:**
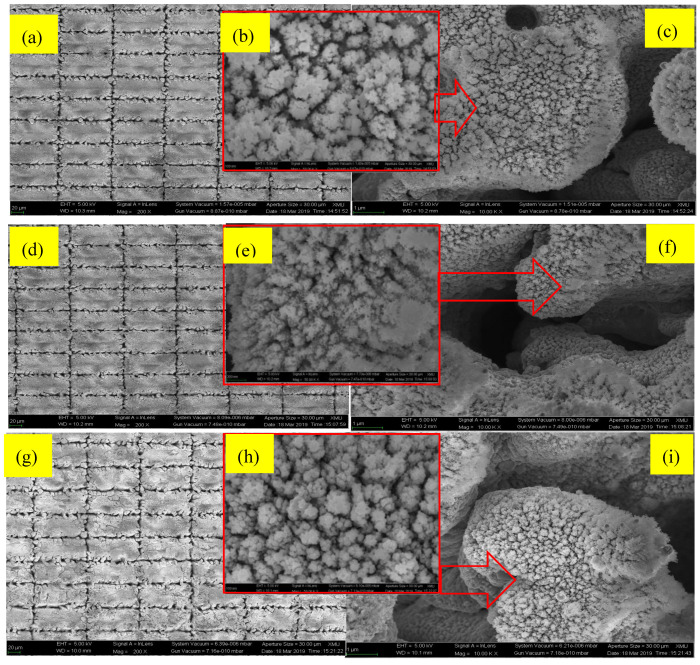
FESEM pictures of the micro/nanostructure morphologies of hydrophobic surfaces where the texturing period was set to 100 μm, and with the following annealing temperatures: (**a**–**c**) 350 °C, (**d**–**f**) 600 °C, and (**g**–**i**) 800 °C. The magnification in (**b**,**e**,**h**) is 50,000. The scale bar of the SEM Figure in the left row is 20 micrometers, the scale bar of the right row is 1 micrometer, and the scale bar of the middle row is 100 nm.

**Table 1 micromachines-11-01057-t001:** Laser processing of polished samples’ metal after thermal annealing, warmed in muffle furnace for 2 h and thereafter cooled in ambient air.

Polished Samples	Laser Processing Results	Microscopy of Metallographic Substrates Surface Structures [24]
Brass sample after 350 °C annealing	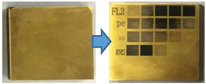	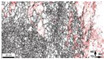 small-sized grains [24]
Brass sample after 600 °C annealing	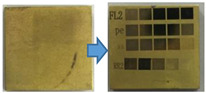	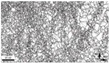 medium-sized grains [24]
Brass sample after 800 °C annealing	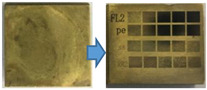	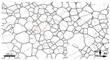 large-sized grains [24]

**Table 2 micromachines-11-01057-t002:** Laser texturing parameters (used in deriving the results presented in Figure 1, Figure 2 and Figure 3).

Tuned Items of Laser	Column 1	Column 2	Column 3	Column 4	Column 5	Other Parameters
Laser fluence (J/cm^2^)	7.16	8.60	10.02	11.46	12.88	Period: 100 μm Scan speed: 1 mm/s Repetition rate: 200 kHz
Period (μm)	80	90	100	110	120	Laser fluence: 8.60 J/cm^2^ Scan speed: 1 mm/s Repetition rate: 200 kHz
Scan speed (mm/s)	0.1	0.5	5	10	100	Period: 100 μm Laser fluence: 8.60 J/cm^2^ Repetition rate: 200 kHz
Repetition rate (kHz)	100	500	1000	2000	5000	Period: 100 μm Laser fluence: 11.46 J/cm^2^ Scan speed: 5 mm/s

**Table 3 micromachines-11-01057-t003:** EDS scans of the micro/nanostructure morphologies of hydrophobic surfaces.

Elements	Wt%	At%	Elements	Wt%	At%
Cu	51.92	36.31	C	5.00	18.51
O	6.55	18.20	Na	2.10	4.05
Zn	33.06	22.48	Fe	0.27	0.21
			Pb	1.10	0.24

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
