# Peer review of "Evolution of a Superhydrophobic H59 Brass Surface by Using Laser Texturing via Post Thermal Annealing"

_micromachines, 2020, doi:10.3390/mi11121057_

Round 1
Reviewer 1 Report
The paper describes nanosecond laser surface micro/nanotexturing of H59 Brass surfaces and posterior annealing. The idea is to find the optimal laser parameters and annealing temperatures to obtain highly efficient super-hydrophobic brass surfaces.
The paper is interesting and appropriate for Micromachines. However, the paper has many typing problems that should be corrected by the authors before I can give an accurate opinion about it.
- The writing language should be improved (Example: in line 51 it is written “remains, still remains”
- What is the pulse duration of the laser used?
- In Figure 1, there is no need to have the photographs of the machines that were used. Also, in the legend of the figure there is a mention to an a), b) and c) which do not appear in the figure.
- Line 116: clarify what is the period in this context.
- Line 120: there is a mention to figure 2 but figure 2 do not exist (only its caption is present).
- In line 121 it is said “In addition, water contact angle measurements were performed to get information about the chemical compositions that caused these wettability transitions.” This is not correct as contact angles do not give chemical compositions.
- Figure 3 should be remade. There is two graphs a) and b) which are not mentioned in the legend and do not seems to be related whit the influence of the scanning speed. The legends inside the bar graphs should be magnified, and the quantities represented in the x and y-axis identified. In line 217 it should be (h) 800°C.
- Some of the previous comments also apply to Figure 4.
- Line 243: There is a mention to figure 10 but there is no figure 10.
- Page 8 and 9 and completely incomprehensible!
Author Response
Dear reviewer,
Thank you for so many pieces of precious advice, after all, the revised version has been improved
(1) In Fig.1, the tool applied in the experiment has been introduced.
(2) The processing periods are 110 micrometers.
(3)I had found out laser scan periods experiment data to draw in Fig.3 and conclude the regular results.
(4) Another question is about the evidence of Cu2O. In my opinion, it is another factor except for the micro/nano morphologies. This reason has been shown in the references [18,20], on the other hand, it is my pity that I can not measure the chemical composition of the samples during the experiments now, the old samples have been out of date. The superhydrophobic surface evolution has been carried out about 3 years ago. In my opinion, the results are credible.
(5) The relation between periods and micro/nanostructures can be optimized in those experiments, on the other hand, it can be analyzed according to Casie-Baxter formula optimized through experiments in 3.2 and Fig3
(6) The linguistic has been improved by the Editspring ’s translation.
Best regard

Reviewer 2 Report
The manuscript titled “Evolution of a Superhydrophobic H59 Brass Surface 3 by using Laser Texturing via Post Thermal Annealing” presents a study the wettability of laser textured brass samples. In particular, the authors focused on the study of the time evolution of wettability and try to relate it to the different morphologies.
Though the topic is really interesting on both a fundamental and applicative point of view, I believe it is not suitable for publication.
The manuscript is not well organized and many unclear expressions are used and the train of thoughts is not understandable. The results about different morphologies obtainable with annealing and laser texturing are not well presented just like the one on wettability.
Access to the manuscript content is made extremely difficult due to layut issues and general carelessness in referring to mislabeled and even missing figures. Also the axes of the plots presented in Figures 3 and 4 have been wrongly labelled.
- Lines 20-21: “The 20 water contact angles (WCA) became larger, and appeared more faster than normal (after an evolution 21 time of 4 days)”; what do the authors mean? I think they refer to the aging behavior of the laser textured surface wettability, but it should be expressed better;
- Lines 30-32: “the distances between the different chemical components in the textured part of the surface, which were observed to be strongly related to the wettability transition time (i.e., from hydrophilicity to super-hydrophobicity), became shortened”; What do the authors mean and which measurements support this conclusion?
- Lines 71-75: “The surface energy, surface roughness and morphology, as well as the dual scaled particles in the micro/nanostructure, play important roles in this evolution procedure (with an exception for the Cu2O nanoclusters )[19,20]. As a matter of fact, the physical profile of the hierarchic micro/nanostructures, in addition to the large oxygen content, will speed up the formation of Cu2O material on the surface.” Do the authors intend that the presence of Cu2O nanoclusters does not influence the evolution to hydrophobicity? If so, why it is important to mention Cu2O?
- I believe the authors should explain with fully described figures the difference between nanoparticle, nanoclusters and nanograins and their formation process? The first paragraph of the paper should be dedicated to the description of the influence of the process parameters on the final morphology.
- The last column of Table 1 is redundant since it described the same procedure for all the temperatures. The authors should include the description of this procedure in the caption of the table.
- Lines 131-133: “These laser fluences have the capacity to change both the brass roughness and the brass oxidation. That is, the wettability will be able to go from hydrophilicity to hydrophobicity in a very short time”; no results are mentioned in this section of the paper to support this statement. This section is dedicated to texturing. Here it should be described how to obtain different morphologies. And then here you should explain roughness and oxidation before and after annealing and before and after the laser treatment.
- In each row of Table 1 textured samples are shown. In each sample four rows of textured areas are visible. The first one should correspond to the samples obtained by varying the fluence (labelled as FL2), the third to the samples obtained varying the scan spee (ss) and the fourth to the sample obtained varying the repetition rate (RR2). Is it rigth? If so, in the manuscript, the authors should correct lines 134-138. What does it represent the textured areas present on the second raw in the sample shown in Table 1? Moreover, the description of “the first row of sample in Table 1”, “the second row of sample in Table 1”, “the third row of sample in Table 1”, besides being wrong because of what I explain above, is also misleading and unclear.
- The mentioned Figure 2 is missing at all.
- Lines 139-140: “The WCAs were measured after the laser processing. In addition, the laser parameters were also optimized”; here I got that you textured the samples after annealing. Then you measured the contact angle. Finally, you optimize the laser parameters. Should not it be the other way round? So, you first optimize the laser parameters and then you measure the water contact angle? It is not clear.
- Lines 140: “From the resulting graphs, including information about relevant nanoparticles,...”; Which graphs do the authors refer to? You should use the number of figure.
- More than once the authors mentioned that the “optimize” the laser parameters. Though, the never explained how, what outcome they check to decide the laser parameters were optimized.
- Wenzel and Cassie-Baxter formulas are mentioned with references.
- Several figures are missing or misplaced in the manuscript, do not have the caption, have the wrong label.
- Lines 162-164: “In addition, when the sizes of the nanograins become smaller, the distances between the nanoclusters will be longer. Hence, Cu2O nanoclusters can be formed quite quickly”; The authors should explain fully this deduction.
- I got the WCA depends on the features of the textured structures; nevertheless, I believe it should be better explain how the depth and the size of the grooves and the size of the grains vary with the fluence and the other laser parameters and only after connect the wettability to the different morphologies. The authos mixed the different topic, thus making difficult to understand the results.
- Lines 190-191: “While the WCA was measured in the first measurement, a textured brass surface, with deeper grooves, were processed by using a higher laser fluence”; it is not clear what the authors would like to explain.
- Line 199: I believe the authors refer to the wrong figures. Should not they be Figure 3 (d), (f) and (h)?
- Could the authors explain what they mean with “hot spots”? Do not they scan the sample continuously? If the sample is scanned continuously, then one can define the pulses per spot, but in each point of the sample the same pulses per spot impinge. If the laser radiation is concentrated to some host spots, the authors should refer to number of pulses which accumulate on the hot spots, rather than the scan speed. The authors should add a sketch for describing the scan strategy.
- Lines 222-226: “Also, the large oxidation degree of the surface can be seen by the color of the brass surface. In the initial tests, a large surface energy was corrected to a more hydrophilic surface. Also, the increase in annealing temperature was found to have a significant effect on both the increase in number of nanograins, and of the hydrophobicity”; it is not clear how the authors infer the oxidation. Moreover, if the authors do not refer to any image, it is quite difficult to understand the effect of temperature on the number of nanograins. Moreover, the authors should perform a quantitative analysis of the nanograins distibutions and size.
Conclusions are partly hide by an EDS spectrum, which is moreover poorly described.
Author Response

(The authors gave the same response as above.)

Round 2
Reviewer 1 Report
In my first review I made some remarks to the paper that were not considered by the authors. For example, no changes were made in figure 1 or the pulse duration of the laser was not introduced. The writing language seems to be even worse than before, (especially the new sentences in yellow introduced in this second version of the paper).
Other comments:
- All 3D bar graphs shown in Figures 2 to 5 are illegible and should be substituted (maybe by line graphs).
- I do not understand why the Cassie Baxter equation was introduced in this version.
- In the legend of Figure 4 there is two times (f).
- Figure 8 has no 8d.
- Figure 9 must be remade and there is no need to show the EDS spectrum.
As I said in my previous review, the paper is interesting but there are many typing problems that should be corrected before I can give an accurate opinion about it. As a result, I do not recommend its publication in the present form.
Author Response
Dear reviewers,
Thank you for your advice! I revised it according to your precious suggests.
First all, i had redrawed the origin 3D map and made another methods to get some clear output maps.
Second,I had paid more attention to revise the description under someone's help.
Third, i deleted the formula and instead of the description ,on the other hand, the EDS figure was deleted.
all of revision had been sign with highlight.

Reviewer 2 Report
I do not believe the the manuscript has been significantly improved.
Figures (in particular Figs 2, 3, 4, 5) should be more soundful. The dependence on fluence (Fig2), on period (Fig.3), on scan speed (Fig.4) and on repetition rate (Fig 5) is not clear both for the quality of the plots and the way how the data are presented.
I would like the authors to change from histograms to plots where the trend of WCA vs fluence, period, scan speed and repetition rate is presented, in order to make the dependence very clear.
At the moment, the axis label are barely visible. Moreover, I believe that while in Figure 2 (b) the considered scan speed is 1 mm/s, in Figure 2 (c) it is 5 mm/s. I believe that changing other working parameters when the authors are trying to describe the influence of fluence is not completely straightforward.
Since the authors ascribed the wettability behaviour also to the morphology, I would like they to put the sistematic study of how the morphology change with the working parameters before showing the trends of WCA. Here, the authors should make clear the scale bar in SEM images, otherwise their explanation about the dimensions of grains could seem just speculations.
I also suggest to add, once suitably amended, the results about the dependency of WCA on the working parameters in Results and discussion section (after the morphological and EDS characterization).
In addition, the authors added Wenzel and Cassie-Baxter equations without explaining their role in describing the wettability, their differences and without incuding references (and without explaining which one is Wenzel and which Cassie-Baxter).
Author Response
Dear reviewer,
Thank you so much for your advice.I had redraw the Fig2,3,4,5 and instead with the 2-D curves with plots. I am so sorry to appliy the 3D express. I will make another improvement.In fact,the scalebar is difficult to show in the Figure, as a result, i mark the scalebar diamension in the new illustraion. The Casie-baxter formula had been instead with the explain. The other hand, because, the samples evolution has been insteaded.
All this revision had been mark with high light words.
best regard
Xizhao Lu

Round 3
Reviewer 1 Report
Dear Editor,
The authors have significantly improved the paper but there are still some issues to be addressed.
The major problem with the paper is related to Figure 1 to 3. The legend inside the figures are to small to be understand. The graphs in Figure 1a, 2a and 3a should be clearly explained: is the data showed corresponding to all experiments made with laser fluence (or period or scanning speed)? what is the meaning of coverage, mid-level and unnormal? What is the significance of the bar size in the legend inside the figures? Also, the legend of these figures (and figure 4) refers 3D bar which are not present anymore.
Line 128: Maybe it´s Figure 1 instead of Figure 2
Line 79: “In addition, water contact angle measurements were performed to get information about the chemical compositions that caused these wettability transitions”. This is not correct as water contact angle measurements do not give information about the the chemical compositions.
Line: 99: It is written “According to the Cassie-Baxter formulation, a microstructure grate will be formed by the laser.” The sentence is not correct as it´s seems that the Cassie-Baxter equation predicts that the laser will produce a microstructure grate.
Table 1 and all over the text: it should be “kHz” and not “KHz”
There is two sub section 4 (Results and Discussion and Conclusions)
Line 313: The sentence “To reduce the laser scan rate, the brass oxidation was extended and the distances between the nanograins were increased” is weird as it seems that brass oxidation was extended and the distances between the nanograins were increased to reduce the laser scan rate.
Author Response
Dear Reviewer,
I correct the mistakes and pay more attention to redraw the maps, and explain the coverage WCA which are plots included in the rectangle bars. The evolution of average WCA will means the speed influenced by substrate heated. Thanks for your warm notice.
yours truly

Reviewer 2 Report
I believe that after some minor amendments (I will describe below), the paper could be accepted for publication.
Section 3: I believe the word "formation" for addressing to the water contact angle is not perfectly right. Perhaps, it would be better to use "variation", "evolution", "modification".
I appreciated the new plots (instead of 3D bars). Nevertheless, I have to say that the label are still not visible enough. Moreover, If I cannot read the axis label or the text into the plot, it is also difficult for me to catch the results of the research. It is a pity because there are interesting results, but they are not suitable presented. I also suggest to update the legends so that it do not have WCA1, WCA2 and so on but rather the values of fluence (Figure 1 (c), (e) and (g)), the values of periods (Figure 2 (c), (e) and (g)), the values of scan speed (Figure 3 (c), (e) and (g)), the values of repetition rates (Figure 4).
I also believe that if one wants to hihglight the influence of one particular parameter (e.g. in Figure 1 the laser fluence), it is not necessary to show the trend of the WCA vs time at a fixed fluence. I still do not get why the authors present plots in Figure 1, 2 and 3 (a) and (b), and in fact they never refer to these plots in the text. The useful plots are in Figure 1,2,3 in (c), (e) and (f) and the related surface morphologies are also interesting to be showed, I would remove the plot in (a) and (b) in Figure 1,2,3. They are not sound where they are placed.
The authors correctly removed the Cassie-Baxter formula, but they still refer to it in line 179 "According to Cassie-Baxter and Wenzel formula, like formula (1), ...". The authors should remove the sentence "like formula (1)" but add the reference to Wenzel and Cassie papers. Moreover, please, be sure Cassie is mentioned with two s. In line 99 and 179 "Casie" is rather wrongly written.
In addition, I would suggest to make the scale bar in SEM image more clear. I saw the authors explicitly wrote what the scale bar is, but since it is still barely visible, it is not really useful.
Author Response
Dear review,
Thanks for your kindly advices and I correct the mistakes and pay more attention to redraw the maps to read clearly, and explain the coverage WCA which are plots included in the rectangle bars. The evolution of average WCA will means the speed influenced by substrate heated.
